# Experiences of Sex Workers in Chicago during COVID-19: A Qualitative Study

**DOI:** 10.3390/ijerph20115948

**Published:** 2023-05-25

**Authors:** Randi Singer, Sarah Abboud, Amy K. Johnson, Jessica L. Zemlak, Natasha Crooks, Sangeun Lee, Johannes Wilson, Della Gorvine, Jahari Stamps, Douglas Bruce, Susan G. Sherman, Alicia K. Matthews, Crystal L. Patil

**Affiliations:** 1Department of Human Development Nursing Science, College of Nursing, University of Illinois Chicago, Chicago, IL 60612, USA; 2Ann & Robert H. Lurie Children’s, Chicago, IL 60611, USA; 3College of Nursing, Marquette University, Milwaukee, WI 53233, USA; 4Howard Brown Health Center, Chicago, IL 60613, USA; 5Southside Health Advocacy Resource Partnership, Chicago, IL 60653, USA; 6Department of Health Sciences, DePaul University, Chicago, IL 60614, USA; 7Johns Hopkins School of Public Health, Baltimore, MD 21205, USA; 8Columbia University School of Nursing, New York, NY 10032, USA

**Keywords:** sex workers, qualitative design, COVID-19, health, safety

## Abstract

COVID-19 exacerbated health disparities, financial insecurity, and occupational safety for many within marginalized populations. This study, which took place between 2019 and 2022, aimed to explore the way in which sex workers (*n* = 36) in Chicago were impacted by COVID-19. We analyzed the transcripts of 36 individual interviews with a diverse group of sex workers using thematic analysis. Five general themes emerged regarding the detrimental impact of COVID-19 on sex workers: (1) the impact of COVID-19 on physical health; (2) the economic impact of COVID-19; (3) the impact of COVID-19 on safety; (4) the impact of COVID-19 on mental health; and (5) adaptive strategies for working during COVID-19. Participants reported that their physical and mental health, economic stability, and safety worsened due to COVID-19 and that adaptive strategies did not serve to improve working conditions. Findings highlight the ways in which sex workers are particularly vulnerable during a public health crisis, such as COVID-19. In response to these findings, targeted resources, an increased access to funding, community-empowered interventions and policy changes are needed to protect the health and safety of sex workers in Chicago.

## 1. Introduction

Sex work can be described in a number of ways; however, for the purpose of this study, sex work is defined as the exchange of oral, anal, or vaginal sex for something of material value [1]. Prior to COVID-19, complex structural and social factors negatively impacted sex worker health in the United States (US) [2]. Criminalization, targeted policing, environments that limit worker agency, poverty, violence, discrimination within healthcare settings, and stigma, all limit sex workers’ abilities to practice harm reduction and health promotion [3,4]. There is growing evidence of the deleterious impact that COVID-19 had on sex workers within the US and beyond [5,6,7,8]. In this study, we explored the ways in which sex workers in Chicago were impacted by the COVID-19 pandemic. Initially, this qualitative study aimed to explore the HIV prevention, self-management practices of sex workers in Chicago in order to support responsive HIV prevention and harm reduction programming. Because the study launched in March of 2020, we additionally explored the ways in which sex workers in Chicago were impacted by COVID-19.

Research points to a negative cascade in relation to sex work and the potential for harm. For sex workers, an increased exposure to trauma and structural violence likely increases the prevalence of physical or emotional harm and the potential for the acquisition of sexually transmitted infections (STIs), including HIV [9]. Research additionally shows, however, that sex workers’ experiences of stigma and criminalization perpetuated by systemic inequities inhibit access to protection by law enforcement, increase barriers to financial stability, and reduce the options for trusted healthcare and harm reduction services [10,11]. Consistent with others within marginalized and stigmatized populations, the prevalence and intensity of such experiences were further deepened by the COVID-19 pandemic [12,13].

With the exception of 10 (rural) counties in Nevada housing legalized brothels [14], sex work in the United States is illegal, resulting in health, economic, and safety repercussions for sex workers [10]. The vulnerability associated with the illicit nature of selling sex increases sex workers’ vulnerability to violence [15], with over 75% reporting lifetime experiences of physical or sexual workplace-related violence prior to the pandemic [16]. The illicit nature of sex forces most of this work to the informal economy. During the pandemic, stay at home orders disrupted work within the informal economy. Sex workers who continued to work were not only at an increased risk of COVID-19, but they also experienced financial strain related to the loss of income. This loss of income prompted greater risk-taking and exacerbated experiences of violence beyond that which is typical.

Structural vulnerabilities such as unstable housing, gender inequality, and lesbian, gay, bisexual, transgender, and queer (LGBTQ+) identities increased incidents of physical and sexual violence perpetuated by clients, partners, and by law enforcement related to selling sex and may have increased the risk of exposure to STIs and HIV [17]. Consistent with findings prior to COVID-19, sex workers at the intersection of multiple marginalized identities were burdened by heightened negative outcomes when attempting to access health-related support [18]. Engagement in sex work, coupled with an identification as someone with limited financial and housing resources, or as someone who is not white, cisgender, or heterosexual, correlated with increased negative incidents experienced by sex workers [6]. During the early months of the pandemic, many healthcare facilities stopped in-person visits, limiting access to STI screening and HIV prevention services such as Pre-Exposure Prophylaxis (PrEP).

When COVID-19 government relief became available for the millions unemployed in the US, sex workers were not prioritized as a vulnerable community as their work was unreportable due to criminalization and thus were not financially protected through the program [7,19]. In the United States, informal economy workers, such as sex workers, were ineligible for governmental safety programs therefore increasing the risk of economic instability during the pandemic. Despite the inherent risks, sex workers continued to work during COVID-19. COVID-19 restrictions limited their clientele and decreased the ability to vet clients, thereby increasing vulnerability to violence and sexually transmitted infections [1]. Client-perpetrated violence is associated with an increased risk of HIV infection among sex workers related to threatened condom negotiation and forced or pressured sex [20]. The rates of interpersonal violence skyrocketed during the pandemic, potentially increasing the vulnerability of sex workers to HIV and STIs [7,21].

In addition to increased incidents of violence, the US additionally saw an amplified mental health decline among the general population, such as a rise in depression, anxiety, substance abuse, and suicidal ideation [22]. Prior to the pandemic, sex workers experienced high rates of depression, anxiety, and post-traumatic stress disorder (PTSD). For example, as many as 61% of sex workers screened positive for PTSD prior to the pandemic, similar or worse scores than combat tour veterans [23]. With the huge upheavals in income, experiences of violence, and lack of access to care during the pandemic, the mental health of sex workers may have further worsened.

### Intersectionality and Structural Violence Framework

Our interview guide used an intersectional lens to highlight the structural inequities experienced by sex workers. Intersectionality has been pivotal in understanding the experiences of marginalized individuals, as the intersections of gender, race, sexuality, and socioeconomic inequality are closely related to the health of sex workers [6]. The combination of multiple disadvantaged identities and social positions is relevant to adverse health outcomes and is crucial in understanding the impact of COVID-19 among sex workers in Chicago.

## 2. Method

### 2.1. Community Empowerment

This qualitative study was guided by a community empowerment framework [24]. Community empowerment is a collective process where both individuals and groups are actively involved in the process of change. In order to shift power from the institutions to the people impacted, interventions are designed, implemented, and evaluated by the community served [25]. This process highlights the consumers of future interventions as experts in their own lives and leans on lived experiences, cultivated knowledge, and collective leadership to inform services and interventions [26,27]. Community empowerment aims to elevate social support and increase individual, financial, and community resources for sex workers in an effort to simultaneously decrease vulnerability and structural barriers to needed care [28]. The centering of community voices inherent to this study is in direct alignment with the community empowerment framework. For this reason, we ensured that sex workers and community partners were an integral part of the research team from conception through dissemination. Study protocol, funding mechanisms, and interview guides were established in partnership with sex workers.

### 2.2. Procedure

We conducted a qualitative data (*n* = 36) analysis aimed at the development of community empowerment interventions by understanding the self-management, health promotion, and harm reduction needs of sex workers in Chicago [8,29]. The study took place from 2019 to 2022 and was approved by the institutional review boards of University of Illinois Chicago.

Participants were recruited through flyers posted in a Chicago-based clinic, on social media (i.e., Twitter, Facebook, and Instagram), and a private community list-serv and word-of-mouth. Potential participants reached out to the study team and were evaluated for eligibility. To be eligible, participants had to fulfil the following: (a) be 18 or older; (b) have exchanged oral, vaginal, or anal sex for something of value in the past 12 months; (c) live in the Chicago area; (d) speak and understood English; and (e) were willing and able to provide informed consent. Once informed consent was obtained, a one-on-one interview was scheduled and conducted over a secure institutional Zoom platform. Current and former sex workers were trained as qualitative interviewers and followed a semi-structured interview guide that was developed in collaboration with community stakeholders. The interview guide covered topics related to physical, sexual, and emotional health, experiences with healthcare, HIV/STI prevention, harm reduction techniques, and the impact of COVID-19 on health, work, finances, and safety. Examples of questions include the following: “how has this pandemic impacted your work?” and “how has this pandemic impacted your safety?”. When determining a sample size for qualitative interviews, it is necessary to balance code saturation (enough codes so that researchers have “heard it all”) and meaning saturation (having enough data to understand participants’ experiences). We reached saturation in our findings at 20 participants; to ensure a diverse sample, we included more participants. Thus, our final sample size of 36 was adequate to answer our research question [30,31,32].

### 2.3. Data Analysis

Interviews were professionally transcribed verbatim and then reviewed for accuracy. Each transcription was deidentified and assigned a unique number and pseudonym. We used Dedoose to code transcripts and thematic analysis to identify codes and common themes [33,34]. Because the objective of this study was to explore the impact of COVID-19 on sex workers, we coded only the data relevant to determining the ways in which COVID-19 affected the overall health, economic stability, and safety of the participants. The second author and a research assistant coded the first 14 transcripts to ensure the validity of the coding and to discuss the dominant thematic categories. We met regularly to discuss the coding scheme and develop the codebook. Once the codebook was finalized, 2 additional co-authors (the 4th and 5th co-authors) coded the remaining 22 interviews. We then identified recurring patterns and themes across the data, based on common categories related to the impact of COVID-19 on physical heath, mental health, economic stability, and safety. We met regularly to discuss the data analysis process.

### 2.4. Research Trustworthiness and Rigor

We followed Morse’s criteria to ensure study rigor credibility, confirmability, and transferability [35]. To establish trust with the participants, trained interviewers were additionally community champions. The one-on-one interviews provided rich descriptions of the participants’ experiences. Authors RS, NC, JS, and community champions LV and MV conducted the interviews, following the same interview guide for consistency; they debriefed and discussed initial findings after the interviews. A coding system was developed to analyze the data and to ensure intercoder reliability, based on three interviews through a consensus approach [36]. We ensured transparency in our findings by including rich participant quotations in our results. Credibility was ensured through prolonged engagement with the data (i.e., multiple readings of transcripts and discussions of meaning, context, and quote selection [35]). The final sample size supported the data of the study. We kept an audit trail that consisted of our memos, observations, and data analytic notes.

Regarding positionality, our team includes former sex workers, community partners, healthcare providers, researchers, and students. Our team consists of individuals who identify as Black, Asian, Arab, LatinX, and white and have various sexual and gender identities. We additionally created a community advisory board to provide feedback on study materials (i.e., recruitment and interview guides) and inception. We have complementary expertise in minority health, mental health and sexual health, health disparities/equity, and qualitative research. Our multiple identities and expertise provide insider and outsider perspectives and influence our research interests, including the approach of this study. To address our biases, we conducted regular meetings, debriefed, and discussed the research process, findings, and implications to ensure confirmability and the rigor of the study. Direct quotations from participant interviews have been incorporated into the narrative. A sufficient description of the methodology and contextualized narratives of participants supported the transferability of the findings. Pseudonyms were used to protect participant confidentiality.

## 3. Results

### 3.1. Participants’ Demographics

Participants were on average 32.71 (±7.6) years old. Participants were racially and ethnically diverse, with the majority being either Black (*n* = 16, 44%) or white (*n* = 9, 25%). Twenty-three participants identified as queer, gay, bisexual, or pansexual, and ten identified as straight. Less than half of the participants identified as cisgender women (44.4%). We provide additional details on the demographic characteristics of the participants in Table 1.

Five themes were identified showing the impacts of the COVID-19 pandemic on sex workers in Chicago: (1) physical health; (2) mental health; (3) economic impacts; (4) safety; and (5) adaptive strategies to new working conditions.

### 3.2. Physical Health

We defined physical health as experiences related to the impact or threatened impact of COVID-19 infections on participants and their clients or personal contacts related to selling sex. Under this theme, and as described in the following sections, we identified two sub-themes: (1) *Self-perception of COVID-19 risks* describes perceived vulnerabilities from sex work to physical health implications (i.e., fever, difficulty breathing, or lethargy) of COVID-19 infections and (2) *Behaviors if participants developed COVID-19 symptoms* describes actual or potential behaviors participants might have employed if they were exposed to or infected with COVID-19.

#### 3.2.1. Self-Perceptions of COVID-19 Risk

Many participants were concerned about how their job increases the risk of acquiring COVID-19 and how becoming infected would impact their and others’ health. In response to physical health concerns, participants decreased their in-person services to avoid exposure to COVID-19. While some participants shifted to online services, others felt forced to risk their physical health due to financial pressures. Nearly a quarter of participants (*n* = 8) provided only virtual services (i.e., OnlyFans, sending nude pictures, and Facetime). Participants remarked that they could not control the degree to which their clients were taking precautions, making in-person services inherently risky. Tracy (white cis woman, 34 years old) described the measures she took:


*I have stopped seeing clients. I stopped booking in person. Totally. They’re still trying to book me now. And I told them repeatedly, it’s not safe to book right now. I’m not booking right now, you’re going to have to wait, but it bothers me that these people are being this sort of laissez faire about this…*


However, while some participants and their clients strictly adhered to the shelter-in-place order, over two-thirds continued some level of in-person services. Participants reported that continuing in-person services was a risky choice they felt forced to make due to financial pressure. This raised fears of being exposed to COVID-19, as well as fears of inadvertently exposing their clients, as illustrated by Jasmin (white trans woman, 30 years old):


*I am concerned that, you know, every time I go out and see a client. I don’t know where they’ve been. I don’t know what they’ve been doing… Like, what if I’m asymptomatic and you know, they get it. And what if they f**king either die, or they give it to like their partner… I can’t have that on my conscience.*


Some participants felt that they were at a low risk of contracting COVID-19 or developing severe complications and thus did not see a need to curtail in-person services. Participants, however, still noticed a decrease in clients because many of their clients were themselves sheltering in place to avoid exposure. Ultimately, almost all participants acknowledged that in-person sex work carried a high risk of COVID-19 exposure due to the close physical proximity it required, and they attempted to adapt accordingly. Krystal (Black cis woman, 41 years old) shared the following:


*I am aware of the risk, and that’s a risk I take when I decide that I would like to indulge in some adult activity, but it is not a fear, you know what I mean? Like a constant fear. I understand you got to be cleaning your hands and you got to have your face mask and all that good jazz and stuff…. It slows up the clientele a little.*


#### 3.2.2. Behaviors if Participants Develop COVID-19 Symptoms

Almost all participants stated that they would adhere to isolation guidelines if they were to develop COVID-19 symptoms or test positive. Many said that they would seek medical treatment at a hospital or doctor’s office for COVID-19 symptoms. Some participants reported that they would try to inform close contacts, including clients. Mayra (Latinx cis female, 25 years old) described how she would inform everyone she was in sexual contact with if she were to test positive:


*I would quarantine right away. Yeah, and inform everyone who I had interacted with … as soon as I had those symptoms. ‘Hey, I’m having these symptoms, maybe you should quarantine or seek your, you know… your health care provider to let them know that you want to get tested.’ I would get tested for antibodies right away.*


### 3.3. Mental Health

The mental health theme describes experiences of mental health symptoms or symptom management, during the early stages of the COVID-19 pandemic. Many participants described the toll the pandemic took on their mental health, while others described the strategies they used to manage stress. Participants named multiple contributing factors that affected their mental health, including financial insecurity, COVID-19-related health concerns, safety concerns, and social isolation. They reported increased anxiety and stress about how lost income would impact their ability to pay for rent, bills, or groceries. Lauren (white cis woman, 30 years old) shared her perspective:


*I’m definitely more stressed. Definitely experiencing more stress. Not just related to my job, but just worrying about family and worrying about my physical health, and like, am I going to get sick. And if I do get sick, how long am I going to be sick for… what kind of toll is that going to take on my…financial security. And, you know, we know that in a capitalist society…if your financial security is not good it very quickly affects every other area of your life.*


Some participants spoke about the interplay between the pandemic and their pre-existing mental health conditions. For example, some participants increased their use of drugs and alcohol to cope with pandemic-related stressors, triggering concerns about their addiction history. Joy (multiracial cis woman, 26 years old) described the impact of the pandemic on her mental health and her use of drugs and alcohol as follows:


*I think a big thing too is that obviously just it’s a stressful time; it’s depressing, it’s isolating, and I’ve noticed that that affects the way that I use drugs and alcohol. And so, I think like I’ve definitely been using a lot more. I’ve had a couple of nights where, you know, I’ve been a little bit too heavily and I know with my own history of addiction and depression that it’s a slippery slope for me. And so, I think that’s one way that it’s definitely affected it.*


On the other hand, a few participants considered that their pre-existing mental health conditions prepared them to better cope with the pandemic. For example, Johnson (Black cis man, 39 years old) stated that he was already used to the social isolation caused by the pandemic because he would already often isolate himself due to depression:


*I was already… into so much fog and depression that I was comfortable with it because I had already been doing it. I was like well, I’m used to this. This is what I’ve been doing. I kind of cut everyone off anyway and kind of isolated myself… It probably brought everybody down to the level I was already at.*


Multiple participants saw the pandemic as an opportunity to address mental health issues that they previously could not. One participant thoroughly discussed how the pandemic afforded them more time to practice self-care to cope with stress and trauma. They even mentioned that being unable to carry out sex work was beneficial to their mental health because of the stress associated with that work. Sam (white genderqueer person, 31 years old) recalled the following:


*I felt like, especially during the early months, it felt like the whole emotional roller coaster within a day, it felt like months of emotions compressed into a day…. But I think that losses that I had in my life have been coming up now that I have time to process them, like deaths of loved ones. I think it’s helping to process it. I don’t think I had time before. So [I’m] actually really grateful for having had the time to feel grief again, because then how else would you like finally move on and like evolve it into a different thing.*


Some participants expressed that the social isolation caused by the pandemic was particularly harmful to their community’s mental health and stressed the importance of peer support networks in this community. Billy (Black cis man, 35 years old) stated as follows:


*Well, we deal with a lot of people who deal with mental health in this community, so it’s been a struggle. It’s definitely been a struggle to not have that interpersonal connection with people to be able to talk some things through and process some things.*


Finally, some participants discussed the secondary impacts they experienced due to the mental health challenges faced by others around them, both fellow workers and clients alike. Some expressed positive feelings about the role they could play in helping others process and manage their stress, anxiety, and depression, whereas others found this role to be overwhelming and burdensome. Rosa (Latinx cisgender female, 26 years old) acknowledged the ways in which sex work has impacted her mental health:


*I have to take very good care to understand like obviously sex work is very can be traumatic. It can be uncertain and it can be very unstable and that can contribute to a lot of things anxiety, depression, a whole list of mental health issues. And so that’s, you know, really important for me to check in with those.*


### 3.4. Economic Impact

This theme describes sex workers’ experiences navigating financial issues during COVID-19. The participants were no exception to the financial hardships the COVID-19 pandemic created for most people. Nearly all participants reported that their income either decreased (*n* = 28) or dropped to 0 (*n* = 5) during the pandemic, mostly due to a decreased demand for services. Demand was impacted not just by clients’ COVID-19 exposure concerns, however additionally because clients were experiencing financial hardship themselves and thus were unable to afford sex work services. Johnson (Black cis man, 39 years old) described this ripple effect as follows:


*But as a sex worker, it makes it very difficult, again, because business is slow, and people don’t have money to spend and splurge on stuff like that as they used to because they’ve all been laid off. So, it’s just a domino effect across the board.*


Some participants stopped working altogether after deciding to completely cease in-person services. They reported using food pantries, food stamps, rental assistance, and other forms of financial assistance to stay afloat. Due to income insecurity, participants often felt forced to choose between their various expenses, such as housing, phone bills, groceries, health insurance, and transportation. One participant, Pamela (Black cis woman, 28 years old), spoke about this dilemma, additionally noting that the government stimulus did not prioritize her financial burdens:


*In particular, this last month, I had to choose between my cell phone bill and buying groceries, and so…for the first time since I was like 19, my cell service ended up getting suspended because I just couldn’t give them money that I don’t have. And for the last two months it’s been overdue, you know, even savings being depleted […]. When the pandemic started, I was paying for health insurance out of pocket, and it was about $170 a month. So, when it started, I decided I’m going to have to unsubscribe from so many things. And I weighed the options and I decided that I couldn’t really at this point afford to pay $170 something dollars a month for health insurance.*


Another participant, Anna (Latinx cis woman, 33 years old), additionally described the loss of income security when she said the following:


*I have no income security. I have not been able to work, and I went from making, like $2000 a month to $3000 a month to like nothing.*


### 3.5. Safety

We address the safety theme by illustrating the unique impact of the pandemic on the participants’ legal and physical safety. This theme includes a persistent fear of financial exploitation, sexual safety, and legal danger.

Participants were fearful of clients attempting to exploit them by requesting sexual activities beyond what was initially agreed upon. Participants navigated these fears in a variety of ways, balancing their financial realities with their sense of personal safety. Joy (multiracial cis woman, 26 years old) shared her perspective as follows:


*A lot of these men… They think that we’re desperate right now. And so, they’re trying to take control [of] the situation and that’s what scares me a lot too. A lot of these [people] that have never done sex work before that are getting into it at the worst, most dangerous time because they’re so vulnerable. These men will take advantage of them because they know that people are desperate right now.*


Angie (Asian cis woman, 34 years old) described how important it was for her to maintain her sense of choice and agency, despite a client attempting to exploit her financial vulnerability, even if it meant losing an opportunity for income:


*I remember this one guy, he keeps messaging me and he’s like, ‘well, I know it’s hard for providers now. So, let’s do a webcam thing.’ And I’m like, ‘no, I don’t do.’ I try to be nice about it. I’m like, ‘I’m sorry, but I don’t do webcam stuff or online things but if you want to see me…’ And he’s like, ‘well, no’ and then like every week, ‘Hey, let’s do a webcam thing.’ Like he said in a way where it’s like, ‘you don’t have a choice.’ I’m like, ‘excuse me, I do have a choice and you’re not telling me that I don’t have a choice.’ And then I’m like, I’m never going to see this guy again because he’s just making me uncomfortable.*


The nature of sex work places sex workers at a high risk of HIV/STI compared to people in other occupations, given the high coital frequency with different sexual partners. Yet, many participants expressed that their sexual safety was compromised during the pandemic, largely because local clinics either decreased in-person appointments or diverted resources toward COVID-19, limiting the availability of HIV/STI testing and PrEP for workers and their clients. For example, Britney (Black cis woman, 31 years old) described her challenges with PrEP as follows:


*In terms of testing, I was on PrEP and then kind of fell off of it, during COVID. So I haven’t gotten tested in maybe six months. Which doesn’t feel great but I just… When I was on PrEP, I had to get tested every three months and so that was a good routine. So I think COVID has really gotten in the way.*


Some participants reported an increased risk of legal danger. Due to the scarcity of trusted regular clients, the fear of encountering undercover police officers posing as clients was a looming concern. Chris (Black cis man, 46 years old) described it as follows:


*I have been forced to see a lot of new people, potentially exposing me to more cops.*


### 3.6. Adaptive Strategies to New Working Conditions

This theme describes the resilient coping and adaption strategies utilized by sex workers to adjust to new working conditions, and it includes the following: the shift to online platforms, the employment of COVID-19 risk reduction measures, and the adjustment of prices to meet the new market needs.

#### 3.6.1. Shift to Online Platforms

Most participants (*n* = 23) shifted toward some form of online services during the pandemic, such as OnlyFans. Physically, this was a safer alternative to maintain or supplement their income while eliminating physical contact with their clients. Less than a quarter (22%) of participants reported engaging exclusively in online work during the pandemic. Britney (Black cis woman, 31 years old) said the following:


*I had an OnlyFans in May and then I just restarted it yesterday… I don’t really [like] doing online work but you have to with this pandemic, everything is remote now.*


Participants often expressed that shifting to online platforms was part and parcel of the general workforce. For many, this new online territory was a last resort. They expressed that online services were less lucrative than in-person services, in part because they had to establish new clientele. Others felt daunted by the time, energy, and money needed to invest in learning how to get started with online work. Jasmin (white trans woman, 30 years old) shared the following:


*“[Online work is] so difficult, and for, like, so much less pay, but it’s like… but you’re like, like working 40, 50 h a week and like for a while you’re not making anything.”*


On the other hand, a few participants expressed that working online was enjoyable and lucrative, and that they would consider maintaining this avenue of work well into the future. Arden (Black genderfluid person, 32 years old) shared the following:


*Actually, COVID really exposed me to the beauty of online sex work.*


While working online did reduce the COVID-19 exposure risk, it additionally presented other safety challenges that participants had to learn to navigate. For example, some participants expressed a reluctance to work online because they felt it decreased their anonymity and subjected them to increased stigma. Joy (multiracial cis woman, 26 years old) shared her concerns as follows:


*I know a lot of people are moving to online work. But as somebody who’s not fully out as a sex worker and I like to be more discreet about my status, I don’t really have that option.*


Others expressed uncertainty about how to navigate the internet legally and securely, specifically mentioning the impact of the Stop Enabling Sex Traffickers Act (SESTA) and the Fight Online Sex Trafficking Act (FOSTA). These laws, which enable digital platforms to be held liable for third-party content depicting underage or nonconsensual sexual activity, have made risk-reduction, safety, and economic stability far less obtainable for online and in-person sex workers [37]. Pamela (Black cis woman, 28 years old) mentioned the following:


*SESTA/FOSTA and the new laws that came about, make it harder for people like me to protect ourselves… That’s what gets in my damn way.*


#### 3.6.2. COVID-19 Risk Reduction Measures

Participants who chose to continue in-person services utilized several COVID-19 risk-reduction measures. These included requiring testing prior to a session, checking clients’ temperatures, utilizing disinfectant spray, requiring masks and other personal protective equipment, and using hand sanitizer before, during, and after a session. Some participants were strict about their requirements, whereas others were concerned that they would have to negotiate with clients around these measures. Hope (Black trans woman, 29 years old) described the measures she employed as follows:


*So, every time I get in a car with someone, or even someone comes to mine, I always go through the COVID procedure. This person got their mask, you keep your mask on while I’m giving you service. We’re using hand sanitizer wipes, we using hand sanitizer. I frequently take two bottle of hand sanitizer with me.*


Finally, location was additionally a significant part of the participants’ risk calculations. Some chose to meet their clients in hotel rooms, believing hotels have higher sanitation standards than other spaces without sanitation requirements. Kit (white genderqueer person, 20 years old) described the following:


*“Yeah, we’ve been meeting at like in a hotel rooms, which feels like pretty clean. Just because they are cleaned every day, so that feels like safe.”*


#### 3.6.3. Adjustment of Prices

Participants adapted to the pandemic by adjusting their prices. Some participants implemented more flexible pricing for their regulars, in part out of sensitivity to the economic insecurity clients were facing. Most participants reported the need to decrease prices in order to maintain their clientele. For example, Angie (Asian cis woman, 34 years old) described the following:


*When some clients are like, “I’m trying to save up for the future because I don’t know what’s going on. Can I give you how much I can afford?”… Especially if it’s a recurring client, I always say yes. I’m like, “Whatever you can donate is definitely fine.” So, when it comes to the monetary, I definitely have compromised.*


A few participants reported increasing prices due to decreased demand, or due to the greater health risks they were incurring with in-person services, as described by Jane (Multiracial intersex female, 24 years old):


*“However, the business is slow but with supply and demand, prices can go up because now you’re like, I’m charging pandemic rates and I’m taking a risk myself”.*


## 4. Discussion

Our findings revealed five themes related to the impact of COVD-19 on sex workers in Chicago on their physical health, mental health, economic impact, safety, and the adaptive strategies used to work during COVID-19.

The physical health of sex workers, many of whom identify as LGBTQ+ were impacted by COVID-19. Despite higher incidence of STIs and HIV, previous experiences of healthcare discrimination and microaggressions prevent sex workers from consistently accessing needed and preventative care [29,38,39]. Additionally, because these populations tend to have existing comorbidities, infection with COVID-19 carries with it graver consequences than that of someone without comorbid conditions [40].

Mental health in the LGBTQ+ communities was impacted by the isolation experienced within the pandemic [22]. Our study sample was very representative of the LGBTQ+ community. Prior to the pandemic, LGBTQ+ people had high rates of adverse mental health outcomes [41]. Research suggests that during the COVID-19 pandemic, mental health outcomes were worsened in this population in relation to the loss of social networks and community support [22]. Possible implications for those marginalized by both sex work and LGBTQ+ identities were that it may have further exacerbated negative mental health outcomes related to the pandemic. These intersecting disparities highlight the need for increased research about evidence-based mental health resources and support for LGBTQ+ sex workers.

Access to healthcare for sex workers was impacted by COVID-19. Participants discussed interruptions in the PrEP care cascade/continuum. Similarly, previous research has reported that financial barriers limited access to dependable care [6,7,42]. Our findings suggest that the COVID-19 pandemic had a direct impact on sex workers’ mental and physical health, wages, available resources, and access to stable and safe housing. Many lost jobs during the pandemic, majorly impacting the global economy as described by previous research [43]. Global research speaks to the detrimental impact of COVID-19 on those working in the informal economy [44,45]. For example, 960 million individuals working in the informal economy reported decreased earnings during the shelter-in-place of the COVID-19 pandemic [44]. The invisibility connected with working in the informal economy and the criminalization of working in a role that is illegal in much of the US have perpetuated a cascade of negative outcomes, and this was exacerbated by the COVID-19 pandemic among sex workers. Prior experiences of stigma discouraged sex workers from accessing needed care. Sex workers were excluded from government programs offering financial support. Without sex work-generated income, previously self-sustaining sex workers were experiencing unstable housing and increased financial burdens such as access to food, transportation, and medicine. Increased risk-taking due to decreased client vetting, reduced fees for services, and increased mental illness were all fostered by the COVID-19 pandemic. Current government policies exist which worsen this dual burden of working within the informal economy and engaging in work that is illicit.

The participants in this study described multiple forms of safety issues, including a persistent fear of financial exploitation, sexual safety, and legal danger. Sex workers are victims of more violence than the general community at large [16]. Police, paying clients, and intimate partners who are not paying clients are additionally regular perpetrators of violence against sex workers [20,46]. Sex workers in our study additionally shared similar experiences of being exploited by paying clients and their constant fear of being exposed to undercover police officers. The long-term impacts of increased violence on their physical and mental well-being must be explored, given the increased likelihood of being physically or sexually assaulted. Furthermore, having little to no recourse against perpetrators of violence due to the criminalization of sex work adds an additional layer of mental health burden. The invisibility and criminalization of those who continue to be victimized highlight how decriminalization will provide sex workers with equitable opportunities and experiences regarding health, safety, and economic circumstances that they are not given under criminalization. The need to protect the health and wellbeing of sex workers is urgent to ensure that sex workers are never placed in a situation as dire as they have been throughout the COVID-19 pandemic, should a different outbreak occur in the future.

### 4.1. Limitations

Our findings need to be interpreted in the light of some limitations. Although recruitment was conducted in-person during outreach as well as online, interviews were conducted only online, using the Zoom platform. Consequently, those without access to smartphones or computers were unable to participate. By limiting participation to those with access to dependable technology, results may have been skewed to those with greater education, housing stability, food security, and health insurance. Additionally, participants were concentrated in a single geographic location. Sex workers in other parts of the country may have had different experiences.

### 4.2. Implications

The participants expressed significant hardship. An increased attention to structural factors impacting the livelihood and well-being of sex workers is needed, especially as it relates to access to mental and physical health services. These findings are useful for designing prevention and protection guidelines for sex workers in the face of the continued health threats associated with COVID-19, recently emerged risks such as mpox, and the increased likelihood of future threats. Improving access to care through community-empowered and culturally safe care modalities such as group care, telemedicine, and mobile clinics is needed. Because community-empowered interventions are designed, implemented, and evaluated by the community served, partnering with current and former sex workers to develop and implement healthcare options responsive to their needs is likely to improve regular engagement with the health system and reduce incidence of HIV [24,25,47,48].

In addition, the decriminalization of sex work has been shown to successfully prevent HIV among sex workers and their partners [49]. The reduction of punitive actions that corresponds with the decriminalization of sex work allows for health, safety, and financial opportunities to consenting adults who trade sex. Protecting the health and wellbeing of sex workers should be a public health priority; the decriminalization of sex work would not only protect the human rights of sex workers, however would additionally foster harm reduction and disease prevention should another public health crisis/outbreak occur in the future [39].

## 5. Conclusions

While stay at home orders related to the COVID-19 pandemic have subsided in the US, our findings support previous evidence that the impact of social and structural factors on the health of marginalized populations such as sex workers has persisted [23,50]. Our findings show that COVID-19 exacerbated previous physical and mental health disparities and contributed to additional economic instability and lack of safety among sex workers in Chicago. The criminalization of sex work forced this work to exist in the informal economy and on the inability to access safety net programs such as unemployment benefits, resulting in potential health-compromising and risk-taking behaviors for survival [45,49]. Decriminalization policies would help formalize and stabilize income related to sex work and protect this marginalized community. In addition, the disruption of in-person mental and physical health service provisions during the pandemic had a grave impact on communities, such as sex workers, who may experience barriers such as stigma in navigating healthcare systems. Healthcare providers, researchers, and policy makers must make the integration of accessible, culturally safe, and community-empowered health interventions such as group care, mobile health, and telemedicine a priority to address health disparities among sex workers during and beyond the COVID-19 pandemic.

## Figures and Tables

**Table 1 ijerph-20-05948-t001:** Demographic Characteristics (*n* = 36).

	*n* (%)
**Age** (mean, *SD*; range: 20–46)	32.71 (7.6)
**Race/Ethnicity**	
Black	16 (44.4%)
Black Native American	2 (5.5%)
Black Asian	1 (2.8%)
Black Latinx	1 (2.8%)
African American	1 (2.8%)
White	9 (25%)
White Latinx	1 (2.8%)
White Asian	1 (2.8%)
Latinx	3 (8.3%)
Asian and Native American	1 (2.8%)
**Sexual Orientation**	
Straight/Heterosexual	10 (27.8%)
Queer	8 (22.2%)
Bisexual/Queer	4 (11.1%)
Gay	4 (11.1%)
Pansexual	4 (11.1%)
Bisexual	3 (8.3%)
Demisexual	1 (2.8%)
Asexual	1 (2.8%)
A bit of everything	1 (2.8%)
**Gender Identity**	
Cisgender woman	16 (44.4%)
Transgender woman	8 (22.2%)
Cisgender man	5 (13.9%)
Gender queer/fluid	4 (11.1%)
Nonbinary	1 (2.8%)
Man	1 (2.8%)
Intersex female	1 (2.8%)

## Data Availability

The data presented in this study are available on request from the corresponding author. The data are not publicly available to protect the privacy of participants.

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
