# Peer review of "Experiences of Sex Workers in Chicago during COVID-19: A Qualitative Study"

_ijerph, 2023, doi:10.3390/ijerph20115948_

Round 1

Reviewer 1 Report

Dear Author/s,

Thank you for your work.

The title of the paper needs to be modified as this one is not matching with the academic style in addressing any paper's title. 

The abstract needs modification to address the main purpose of the paper.

The introduction needs a revision to give the reader the main purpose of the current study, then you need to have a literature review or theoretical framework to address the gap of knowledge before jumping to the method section. This part needs a major revision. 

The sample: you need to give reasons behind choosing this sample.  In addition that the sample size is too small ( N=36) which is not a representative sample. 

in the implication part, the author refers to "New Zealand" as a reference to compare the current study that focused on "Chicago"!! In such a case, I would expect to see a part about cultural differences and determine the similarity they mentioned.   

The results part from pages 4-10, needs a revision to be addressed as main points and then sub-points.  You focused on good points but you still need to restructure this part to allow the reader to understand the main issues in the findings and be able to link it to the conclusion part. 

The conclusion is really weak and needs a revision to match the findings that you addressed earlier in the paper. 

The language needs a professional proofreading. 

The paper needs major revision. 

Thank you 

Author Response

Reviewer 1 Comments

Response to Reviewer/how and Where addressed

The title of the paper needs to be modified as this one is not matching with the academic style in addressing any paper's title.

The title of the paper has been changed to: 

Experiences of sex workers in Chicago during COVID-19: A Qualitative Study (line 1-2)

The abstract needs modification to address the main purpose of the paper.

The purpose of this study, (2019-2022) was to explore the various ways that COVID-19 impacted sex workers in Chicago.

The introduction needs a revision to give the reader the main purpose of the current study, then you need to have a literature review or theoretical framework to address the gap of knowledge before jumping to the method section. This part needs a major revision. 

Thank you for this thoughtful feedback.  We have revised the introduction according to your recommendations. We discuss how our interview guide used an intersectional lens to highlight the structural inequities experienced by sex workers (line 160-176).  Then, we continue to address the gaps in knowledge prior to discussing the methods.

The sample: you need to give reasons behind choosing this sample.  In addition, the sample size is too small (N=36) which is not a representative sample. 

We chose to interview sex workers because this population is disproportionately impacted by HIV.  Because it is a qualitative study, n=36 is an adequate sample size ( Morse, J. M. (2000). Determining sample size. Qualitative health research, 10(1), 3-5. ). When determining sample size for qualitative interviews, it is necessary to balance code saturation (enough codes so that researchers have “heard it all”) and meaning saturation (having enough data to understand participants’ experiences). We reached saturation in our findings at 20 participants, thus our final sample size of 36 was adequate to answer our research question. 

21.       Miles M, Huberman AM, Saldańa J. Qualitative Data Analysis: A Methods Resourcebook. 4th ed. 
SAGE Publications, Inc.; 2020. 

22.       Merriam S, Tisdell E. Qualitative Research: A Guide to Design and Implementation. 4th ed. Jossey- 
Bass; 2016

in the implication part, the author refers to "New Zealand" as a reference to compare the current study that focused on "Chicago"!! In such a case, I would expect to see a part about cultural differences and determine the similarity they mentioned.   

Thank you for this acknowledgement.  We removed our discussion of New Zealand.

The results part from pages 4-10, needs a revision to be addressed as main points and then sub-points.  You focused on good points but you still need to restructure this part to allow the reader to understand the main issues in the findings and be able to link it to the conclusion part. 

We revised and restructured the results and hope that the main issues have been clarified.

The conclusion is really weak and needs a revision to match the findings that you addressed earlier in the paper. 

We strengthened the conclusion.

The language needs a professional proofreading. 

My mentors and colleagues have reviewed this paper prior to resubmission.

The paper needs major revision. 

I appreciate your recommendation and feel this major revision has improved the flow of the paper.

Reviewer 2 Report

I thank the authors for their efforts to draw the attention to the matter of COVID-19 impact upon persons engaged into what is currently classified as an illegal activity in the Chicago area, as a subgroup of the American population affected by the pandemic. The narrative supports the conclusion on the desirability to legalize the work of the sex workers in order to address/decrease their multiple vulnerabilities. I would suggest to strengthen this point and make the link between the criminal status of their activities and the vulnerability more prominent throughout the narrative – to support the conclusion/recommendations. 

A few suggestions below:

53: it would be useful to include in which states (number) this type of work is legal. Perhaps also include a list of other countries with similar population size, where it is legal and not legal, for context and comparison, possibly with a comparison of measures/policies that specifically addressed this group of population. 

75-76: it would be useful to clarify if these persons registered as officially unemployed. Line 345 states that they were able to benefit from rental assistance, food coupons, etc. this suggests that some form of support was provided to these persons, if not through targeted COVID-19 funds, then through other funding schemes – if indeed the case, these should be clarified.

81-83: this correlation is not clear. Has it been confirmed for the Chicago area?

159: should it be 2.4?

238: a general comment – what is the scientific value of direct quotations, particularly with non-normative lexicon?

 345-362: how is this different from other vulnerable populations in the Chicago area or in the USA or beyond?

363: it is worth expanding how safety is related to the research theme, because the title suggests a focus on the health safety.

392: lack of testing opportunities during the COVID-19 pandemic was not sex-worker specific.  

506-507: “Sex workers were excluded from government programs offering financial support” – how was this established? Was there an exclusion clause? Sources?

535: the total number is for the first time mentioned – a suggestion to move up to the beginning / Absract

4.2. Implications section suggests decriminalization of sex work as the main response – this may be made more pronounced or elaborated upon in the conclusion.   

Author Response

Reviewer 2 comments

Response to Reviewer/how and Where addressed

53: it would be useful to include in which states (number) this type of work is legal. Perhaps also include a list of other countries with similar population size, where it is legal and not legal, for context and comparison, possibly with a comparison of measures/policies that specifically addressed this group of population. 

With the exception of 10 (rural) counties in Nevada housing legalized brothels (Macfarlane et al., 2017), sex work in the United States is illegal, resulting in health, economic, and safety repercussions for sex workers (K. H. Footer et al., 2016).

75-76: it would be useful to clarify if these persons registered as officially unemployed. Line 345 states that they were able to benefit from rental assistance, food coupons, etc. this suggests that some form of support was provided to these persons, if not through targeted COVID-19 funds, then through other funding schemes – if indeed the case, these should be clarified.

When COVID-19 government relief became available for the millions unemployed in the US, sex workers were not prioritized as a vulnerable community as their work was unreportable due to criminalization and thus were not financially protected through the program (Callander et al., 2021; Rogers et al., 2021).

81-83: this correlation is not clear. Has it been confirmed for the Chicago area?

Correlation between sexual assault and increased risk of STIs and HIV has been confirmed.

159: should it be 2.4?

YES.  I changed this.  Thanks.

238: a general comment – what is the scientific value of direct quotations, particularly with non-normative lexicon?

Direct participant quotations are a critical component of qualitative methods. These quotes ensure transparency in findings and interpretations of results. A marker of rigorous qualitative methods is rich, thick participant quotations.        21.       Miles M, Huberman AM, Saldańa J. Qualitative Data Analysis: A Methods Resourcebook. 4th ed. 
SAGE Publications, Inc.; 2020.

 345-362: how is this different from other vulnerable populations in the Chicago area or in the USA or beyond?

There is increased vulnerability related to the illicit nature of sex work. Other vulnerable populations such as LGBTQ or persons of color may not have their health implicated in the same way.

363: it is worth expanding how safety is related to the research theme, because the title suggests a focus on the health safety.

We changed the title as per the recommendation of reviewer 1. Our findings address five different themes, and safety is one of them.

392: lack of testing opportunities during the COVID-19 pandemic was not sex-worker specific.  

It is particularly salient among sex workers b/c of the high risk of sex work compared to risks in other populations seeking HIV testing.

506-507: “Sex workers were excluded from government programs offering financial support” – how was this established? Was there an exclusion clause? Sources?

Work that occurs in the informal economy in the United States, such as sex work, is ineligible for safety net programs such as governmental unemployment support. Therefore, sex workers unable to work during the pandemic were particularly vulnerable to economic hardship.

535: the total number is for the first time mentioned – a suggestion to move up to the beginning / Abstract

Great recommendation.  Thank you.  We have included the number in the abstract.

4.2. Implications section suggests decriminalization of sex work as the main response – this may be made more pronounced or elaborated upon in the conclusion.   

Implications section is intended to suggest that decriminalization of sex work is one of the recommendations that would facilitate harm reduction.  We have elaborated on this in the conclusion as well.

Reviewer 3 Report

Thank you for submitting this article for peer review. Describing the experience of a particular group in the Covid-19 period is an important objective. However, the researchers do not provide any new insights. The themes highlighted may be implicit in advance (impact of COVID-19 on physical health; economic impact of COVID-19; impact of COVID-19 on safety; impact of COVID-19 on mental health; and adaptive strategies for working during COVID-19), making qualitative research shallow.  The authors also argue that "the relatively small sample size (N=36) may not be reflective of the SW community at large" raises questions as to whether they are aware of the essence of qualitative research methodology. Qualitative research is designed to describe a phenomenon, not to validate it, so the sample may be small. It should also be noted that the layout of the article is not tidy, and the format is not followed when citing literature sources, quoting research participants, etc.

Author Response

Reviewer 3 Comments

Response to Reviewer/how and where addressed

Researchers do not provide any new insights. The themes highlighted may be implicit in advance (impact of COVID-19 on physical health; economic impact of COVID-19; impact of COVID-19 on safety; impact of COVID-19 on mental health; and adaptive strategies for working during COVID-19), making qualitative research shallow. 

The insights, though not necessarily novel, are imperative for understanding the climate for sex workers in Chicago.  Findings align with other published reports of sex workers elsewhere highlighting a universal experience for sex workers during COVID-19 and its deleterious impact on physical health; mental health, safety, economics; and the adaptive strategies required to work amidst COVID-19. Structural barriers prevented and continue to prevent sex workers from receiving needed access to care and resources.  Findings highlight the need for structural changes before the next big public health emergency omits the needs of sex workers.

Describing the experience of a particular group in the Covid-19 period is an important objective. However, the authors also argue that "the relatively small sample size (N=36) may not be reflective of the SW community at large" raises questions as to whether they are aware of the essence of qualitative research methodology.

Thank you for this feedback.  We agree about your ideas around sampling and removed sample size as a limitation.

Qualitative research is designed to describe a phenomenon, not to validate it, so the sample may be small.

I agree that qualitative research aims to describe a phenomenon rather than validate it.

It should also be noted that the layout of the article is not tidy, and the format is not followed when citing literature sources, quoting research participants, etc.

We have addressed the formatting of this document.

Round 2

Reviewer 1 Report

Dear Authors, 

Thank you for modifying the paper and replying to my comments. 

Reviewer 2 Report

I would like to thank the authors for their review. The updated version represents a major improvement both in terms of content and the language. I sincerely wish and hope that the narrative would contribute to the change of policy concerning this group of the American population. 

Reviewer 3 Report

Thank you for the improved manuscript.